# Knowledge on tuberculosis and utilization of DOTS service by tuberculosis patients in Lalitpur District, Nepal

**Nirmal Gautam**[1]*, **Rewati Raj Karki**[1], **Rasheda Khanam**[2,3]

**1** Department of Public Health, Nobel College of Health Sciences, Kathmandu, Nepal, **2** School of Business, The University of Southern Queensland, Toowoomba, Australia, **3** Centre for Health Research, The University of Southern Queensland, Toowoomba, Australia

* gnirmal655@gmail.com

**Data Availability Statement:** All relevant data are within the manuscript and its Supporting Information files.

**Funding:** Authors did not receive any financial support for conducting this research.

## Abstract

### Background

Tuberculosis is one of the major infectious diseases and is both complex and serious. It is spread from person to person through the air, causing a public health burden, especially in low- and middle-income countries. This study aims to assess the knowledge on tuberculosis and the utilization of Directly-Observed Therapy, Short Course (DOTS) service from the public DOTS centers in Lalitpur district of Nepal.

### Method

A structured questionnaire was used to collect data from 23 DOTS centers in Lalitpur district. Univariate and multivariate logistic regression was applied to assess the knowledge on tuberculosis and utilization of DOTS among people living with tuberculosis.

### Results

Among 390 respondents, 80% of patients had knowledge of tuberculosis and 76.92% utilized the DOTS service from the DOTS center. People of higher age (50–60 years) [aOR; 13.96, 95% CI 4.79,40.68], [aOR; 10.84,95% CI 4.09,28.76] had significantly more knowledge on TB and utilization of the DOTS service compared to the younger group. Additionally, those who completed twelfth class [aOR; 2.25, 95% CI 0.46,11.07] and [aOR;2.47, 95% CI 0.51,11.28] had greater knowledge of Tuberculosis and utilization of DOTS compared to those who had not completed twelfth class. Likewise, compared to urban residents, respondents in rural areas (aOR; 0.51, 95% CI 0.27,0.97) had less knowledge of tuberculosis, (aOR; 0.57, 95% CI 0.32,1) and less chance of utilization of the DOTS service from the DOTS center.

### Conclusion

Approximately one quarter of patients did not have adequate knowledge of tuberculosis and were not utilizing the DOTS service, particularly in younger age groups, people living in a

**Competing interests:** Authors have no conflict of interest.

combined family, with no education, poor economic position, and from rural areas. Findings of this study revealed that some specific programs are needed for enhancing the knowledge and utilization of DOTS, particularly for those patients whose economic situations extended from low to mid range.

## Introduction

Despite the accessibility of effective regimens and treatment of tuberculosis (TB), it has continued to cause instability and debility in people across the globe [1]. Throughout the world, nearly one quarter of the population (1.7 billion) are at risk of TB infection during their lifetime while in 2018 10 million cases were infected and 1.7 million deaths were reported [2]. The highest number of cases were found in South-East Asia (44.4%), followed by the African region (24.8%), and 18% in the Western Pacific region [3]. Although TB is a major health threat to mankind in both developing and developed countries [4, 5], in developing countries, it poses a significant risk of infection due to the differences in demographic (e.g. overcrowded and poor housing), socioeconomic (e.g. low income, education), and cultural factors [6–11]. Moreover, the discrepancies in these factors are not only considered to be a risk of infection and resulting physical instability but also show the complex relationship between tuberculosis and mental illness among TB patients [12–14]. Thus, people suffering from TB are also victims of psycho-social problems, which cause barriers to diagnosis, treatment and control of the disease [15].

Nepal is a developing country where the majority of the population are living with poor socioeconomic conditions [16]. The poor socio-economic conditions have an impact on health and show the burden of infectious diseases, including TB [17]. People facing poorer socio-economic conditions in Nepal are more likely to develop TB, but the barriers to diagnosis may cause extra problems [18]. As a result, the rate of TB is rising very quickly throughout the Nepalese districts [19]. Every year, more than thirty thousand people are infected with TB [20] And, of these, between five and seven thousand TB patients lose their life from what should be a preventable and curable disease (TB) [21]. To manage this issue, the National Tuberculosis Program (NTP) provides comprehensive health management services focusing on early diagnosis, treatment, and control of TB throughout Nepal [22]. Similarly, Directly-Observed Therapy, Short Course (DOTS) service was also implemented in 2001 for controlling, identifying, and standardizing treatment, including proper diagnosis, monitoring, and reporting of TB cases. Despite these immense efforts, the number of new TB cases has increased from 158 per100,000 in 2014 to 245 per 100,000 population in 2018, whereas the TB case identification rate has been gradually decreasing [22–25]. In 2018, about 69000 people were living with TB, but only 32474 cases were reported in the National Tuberculosis Program [25, 26]. Meanwhile, more than 50 percent TB cases missed diagnosis, and treatment [26, 27].

Thus, one of biggest obstacles for controlling the missing TB cases is lack of knowledge and awareness of tuberculosis among patients [27–29]. Moreover, a judgmental attitude towards TB patients prevents them from requesting proper care and treatment [30]. For these reasons, there is a wide gap in the identification, treatment and control of tuberculosis [24]. Thus, knowledge, awareness, and attitude are essential for improving the overall health of patients [30, 31].

Few studies have been conducted in Nepal based on knowledge of TB related to signs and symptoms, source of infection, diagnosis, and curability among the tuberculosis patients and

general population [32–34]. Moreover, such studies have failed to provide comprehensive knowledge on tuberculosis and the utilization of DOTS and its associated factors among the patients living with tuberculosis [30, 35, 36]. Thus, knowledge on TB should be provided to increase the level of awareness and help to change attitudes towards this disease among all people, including TB patients [37]. Therefore, our study assesses the knowledge of TB and utilization of the DOTS service from the DOTS center by TB patients in Lalitpur district, Nepal. In this study, we were particularly interested as to whether the TB patients had adequate knowledge on TB or not and whether or not those patients utilized the complete DOTS service from public DOTS centers. We also examined the socio-demographic factors of knowledge and utilization of the DOTS service. The association between background variables (age, gender, marital status, family type, education, occupation, income, and residence) with knowledge and utilization of DOTS among TB patients would be an important way ensure the correct action for tackling the disease. To address these questions, we evaluated the knowledge on TB and utilization of DOTS using the measure of the TB report. The results of this paper could make a significant contribution at the planning and policy level which would help to improve case identification and health seeking behavior among health service providers and health service users.

## Materials and methods

### Study area

Lalitpur district is the metropolitan city and is a neighboring district of Kathmandu (capital city) with comprehensive facilities including public services (health, education, communication, transportation) most of which are easily available [38]. Subsequently, most International Non-Government Offices (INGOs), Non-Government Offices (NGOs) and charitable organizations are also located there with the aims of providing equitable and assessable basic fundamental services to the unreached, poor, vulnerable, underserved, population [39]. Despite these facilities, Lalitpur district has a large number of TB cases and is unable to deal with them [21]. Therefore, this study was undertaken at Lalitpur district of Nepal in order to identify the knowledge on TB and utilization of DOTS centers among the TB patients.

### Study design, sampling procedure and study population

Institutional based cross-sectional study design was applied to conduct this study in 23 out of 57 public DOTS centers of Lalitpur district, from 15 June 2020 to 17 August 2020. These DOTS centers were selected using the lottery method of a simple random sampling technique, whereas respondents were selected based on purposive (non-probability) sampling technique. Moreover, the study only focused on tuberculosis patients who were taking the anti-tuberculosis drug from the public DOTS center. However, age of the patients below 20 years and above 60 years were not included. Similarly, those patients who were physically weak and unwilling towards the study were not included.

### Sample size calculation and data collection

The total 623 tuberculosis patients were living in Lalitpur district of Nepal. Of them, 390 sample was determined using the single population proportion formula [40] below,

$$\text{n} = \frac{Z^2_{1-\alpha}P(1-P)}{d^2}$$

Where, n is the desired sample size for this study, $Z^2_{1-\alpha}$ is standard value of 95% confidence interval (1.96), $P$ is previous proportion of tuberculosis (41.2%) [20], and $d^2$ is marginal error (5%). Subsequently, by considering the 5% non-response rate the final sample size was 390. After determining the desired sample size, pre-testing was done in order to identify the errors and misleading questions. All these errors were removed after the pre-testing. Then data collection was done using structure questionnaires which covered a wide range of information on demographics. The socio-demographic variables included were: age (20 to 60 years), gender (male and female), marital status (unmarried and married), family type (nuclear and joint family), ethnicity (Janajati, Brahmin), education (no education, one to seventh class, eighth to twelfth, and more than twelfth class), occupation (Labor, Agriculture, Private job, Government), income (poor, middle, high economic position), and residence (urban, rural). Moreover, tools were modified and adopted from previous literature on same subject (knowledge on tuberculosis) [42, 43]. These sets of tools were provided to field enumerators for collecting the data from respondents. While these tools were first developed in English, they were later converted into a Nepali version and provided to the field enumerators for collecting data. Furthermore, this study was carryout during the COVID-19 pandemic periods, thus all precaution measures were applied to conduct the study.

## Data management and statistical analysis

The collected data were entered in a spreadsheet with the appropriate code for each variable of the study, and the R program was used to analyze the data. Both descriptive and inferential statistics were used for formal data analysis. Descriptive statistics were used to identify the proportion of all the background variables while inferential statistics were performed to identify the strength of the associations between outcomes (knowledge and DOTS service utilization) and independent variables. In the inferential statistics, univariate and multivariate logistic regression models were applied. Adjusted odds ratios (aORs), and 95% confidence intervals (95% CIs), were used to identify the relationship between independent and outcome variables, and P values <0.05 were also considered to indicate the level of statistical significance.

## Measure of knowledge on tuberculosis

Knowledge of tuberculosis by patients was measured based on nine questions. These questions were: (1) *Do you know tuberculosis (2) Is tuberculosis is caused by bacteria, (3) Is tuberculosis communicable, (4) What are the sign and symptoms of tuberculosis, (5) Does covering the mouth and nose while coughing and sneezing help to reduce the spread of tuberculosis, (6) Can tuberculosis be prevented by not sharing the utensils with an infected person, (7) Is tuberculosis cured by anti-TB drugs, (8) How long is the treatment for tuberculosis, (9) Can defaulter and relapse TB be cured.* The response to these questions was either "Yes" or "No". Based on these measures we calculated the overall knowledge on tuberculosis in a dichotomous form. The knowledge on tuberculosis was the outcome variable of this study. The outcome variable was coded "0" and "1". "0" for inadequate knowledge and "1" for adequate knowledge although some literature assessed the knowledge on tuberculosis using the mean score of tuberculosis. This mean score was set a cutoff point; if the score was higher than the mean was considered to be adequate knowledge while lower than the mean value was considered to be poor knowledge on tuberculosis [41, 42].

## Measure of utilization of DOTS service

Utilization of the DOTS service was also an outcome variable of this study and was used to assess whether or not patients completely followed the intensive phase (two months) and the continuation phase (four months) [43]. However, if the patients missed at least one occasion

in the intensive phase or the continuation phase, were not regarded as fully utilizing the DOTS service. Those patients who visited the DOTS center without absenteeism during the intensive treatment period and the continuation period was considered as utilizing the DOTS service from the DOTS center.

### Ethical approval and respondents' consent

Ethical approval was obtained from the Institutional Review Committee of Novel College of Health Sciences, Sinamangal Kathmandu, Nepal (Registration number; 322/2020). The selected respondents were voluntarily asked to participate in the study by procuring their verbal and written informed consent. All the possible risks, benefits, advantages, and confidentiality were fully explained before accepting them in this study.

## Results

Table 1 shows the background characteristics of knowledge about tuberculosis by age, gender, marital status, ethnicity, family type, education, occupation, income, and residence. A total of 390 respondents were recruited from 23 DOTS centers in Lalitpur district. Of these, the majority (80%) of patients had adequate knowledge on TB. This higher percentage (86%) of knowledge on TB was found at the age of 30–39 years. Similarly, male (84.2%), married (81.4%), and patients living in a nuclear family (90.5%) were found to have higher knowledge on TB. In addition, those patients who were of BRAMIN ethnicity (82.7%), completed twelfth class (85%), and worked in the private sector (91.2%) also had greater knowledge of TB. Subsequently, patients from high economic positions (93.1%) and from urban areas (85.1%) had greater knowledge of TB than who were from a poor background and/or a rural area.

Table 2 shows the characteristics of the utilization of the DOTS service by age, gender, marital status, religion, education, occupation, family type, and residence. It was found that the majority (76.92%) of tuberculosis patients have utilized the DOTS service from the public DOTS center. This higher prevalence of utilizing DOTS was found in male patients (79.5%) while 19.5% was found in female patients. Additionally, those patients who were married (87.3%), living in a nuclear family (87.3%) and higher age (40 to 60 years) were also found to have a higher proportion of utilizing the DOTS service. Subsequently, patients from BRAMIN ethnicity (78.8%), working in a government office (88.6%) and living in urban (81.7%) areas were also found to have higher utilization of DOTS service.

Table 3 shows the factors associated with knowledge of TB with age, gender, marital status, ethnicity, family type, education, occupation, income and residence of people living with TB. Among these variables, age, family type, occupation, income, and residence were statistically significant ($< 0.05$) with knowledge of TB. It was found that patients in the 50–60 years age group (aOR; 13.96, 95% CI 4.79,40.68) and those living in a nuclear family (aOR; 3.92, 95% CI 1.79,8.63) were more likely to have knowledge of TB as compared to patients in other age groups and living in joint families. Similarly, patients who were government employees (aOR; 3.75, 95% CI 1.04, 13.15) and had higher economic positions (aOR; 4.25 95% CI 1.14,12.87) had significant knowledge of TB. Furthermore, patients from rural areas (aOR; 0.51, 95% CI 0.27,0.97) were less likely to have adequate knowledge of TB than those from urban areas.

Table 4 shows the factors associated with unitization of the DOTS service among people living with TB. In the multivariate analysis, age, family type, occupation, income and residence were found to be statistically significant ($< 0.05$) with utilization of the DOTS service. Patients from 50–60 years of age (aOR; 10.84, 95% CI 4.09,28.76) were more likely to utilize the DOTS service from the public DOTS center. Likewise, patients living in a nuclear family (aOR; 2.94, 95% CI 1.49,5.82), working in a private organization (aOR; 2.39, 95% CI 1.26,4.54), and

**Table 1. Knowledge of tuberculosis by socio-demographic factors.**

| Variables | Knowledge on Tuberculosis by sociodemographic | |
|---|---|---|
| | Inadequate knowledge (N = 78, 20%) | Adequate knowledge (N = 312, 80%) |
| | N (%) | N (%) |
| **Age** | | |
| 20–29 years | 30 (53.6) | 26 (46.4) |
| 30–39 years | 15 (14) | 92 (86) |
| 40–49 years | 22 (14.4) | 131 (85.6) |
| 50–60 years | 11 (14.9) | 63 (85.1) |
| **Gender** | | |
| Male | 34 (15.8) | 181 (84.2) |
| Female | 44 (25.1) | 131 (74.9) |
| **Marital** | | |
| Unmarried | 21 (25.3) | 62 (74.7) |
| Married | 57 (18.6) | 250 (81.4) |
| **Family type** | | |
| Joint | 63 (27.2) | 169 (72.8) |
| Nuclear | 15 (9.5) | 143 (90.5) |
| **Ethnicity** | | |
| Janajati | 42 (23.1) | 140 (76.9) |
| Brahmin | 36 (17.3) | 172 (82.7) |
| **Education** | | |
| Illiterate | 35 (19.2) | 147 (80.8) |
| Grade 1 to 7 | 37 (21.1) | 138 (78.9) |
| Grade 8 to 12 | 3 (23.1) | 10 (76.9) |
| More than grade 12 | 3 (15) | 17 (85) |
| **Occupation** | | |
| Labor | 50 (34.2) | 96 (65.8) |
| Agriculture | 8 (29.6) | 19 (70.4) |
| Private job | 16 (8.8) | 166 (91.2) |
| Government job | 4 (11.4) | 31 (88.6) |
| **Income** | | |
| Poor economic position | 40 (22.6) | 137 (77.4) |
| Middle economic position | 33 (23.4) | 108 (76.6) |
| High economic position | 5 (6.9) | 67 (93.1) |
| **Residence** | | |
| Urban | 35 (14.9) | 200 (85.1) |
| Rural | 43 (27.7) | 112 (72.3) |

patients from high economic position (aOR; 4.2, 95% CI 1.55,11.38) had a higher chance of utilizing the DOTS service. Patients from rural areas (aOR; 0.57, 95% CI (0.32,1) were less likely to utilize the DOTS service from public DOTS centers than those from urban areas.

## Discussions

Knowledge of TB is one of the key components for diagnosis, treatment, control and elimination of TB. It helps to reduce the risk, remove barriers and efficiently complete TB modules. However, in some corners of the world including Nepal, TB is still recognized as a social disease that carries huge social stigma and discrimination. The misperception towards TB may cause barriers to access knowledge of TB and may also restrict the utilization of the DOTS

**Table 2. Utilization of DOTS by socio-demographic factors.**

| Variables | Utilization of DOTS | |
|---|---|---|
| | No (N = 90, 23.07%) | Yes (N = 300, 76.92%) |
| | N (%) | N (%) |
| **Age** | | |
| 20–29 years | 31 (55.4) | 25 (44.6) |
| 30–39 years | 19 (17.8) | 88 (82.2) |
| 40–49 years | 27 (17.6) | 126 (82.4) |
| 50–60 years | 13 (17.6) | 61 (82.4) |
| **Gender** | | |
| Male | 44 (20.5) | 171 (79.5) |
| Female | 46 (26.3) | 129 (73.7) |
| **Marital** | | |
| Unmarried | 24 (28.9) | 59 (71.1) |
| Married | 66 (21.5) | 241 (78.5) |
| **Family type** | | |
| Joint | 70 (30.2) | 162 (69.8) |
| Nuclear | 20 (12.7) | 138 (87.3) |
| **Ethnicity** | | |
| Janajati | 46 (25.3) | 136 (74.7) |
| Brahmin | 44 (21.2) | 164 (78.8) |
| **Education** | | |
| Illiterate | 42 (23.1) | 140 (76.9) |
| 1 to 7 class | 42 (24) | 133 (76) |
| 8 to 12 class | 3 (23.1) | 10 (76.9) |
| More than 12 class | 3 (15) | 17 (85) |
| **Occupation** | | |
| Labor | 51 (34.9) | 95 (65.1) |
| Agriculture | 8 (29.6) | 19 (70.4) |
| Private job | 27 (14.8) | 155 (85.2) |
| Government job | 4 (11.4) | 31 (88.6) |
| **Income** | | |
| Poor economic position | 44 (24.9) | 133 (75.1) |
| Middle economic position | 40 (28.4) | 101 (71.6) |
| High economic position | 6 (8.3) | 66 (91.7) |
| **Residence** | | |
| Urban | 43 (18.3) | 192 (81.7) |
| Rural | 47 (30.3) | 108 (69.7) |

service from the DOTS center [27, 44–48]. Therefore, this study highlighted the knowledge of TB and utilization of the DOTS service among TB patients in Lalitpur district of Nepal. Studies revealed that approximately 80% of TB patients had knowledge of TB, and the majority of these were from younger age groups (29–38 years). These groups of patients were able to explain signs and symptoms, causative agents, mode of infection and duration of treatment of TB. However, one fifth of patients did not have adequate knowledge on TB and were unable to explain it. The findings of our study indicate that inadequate knowledge of TB would result from poor health seeking behavior, and would cause delays in case findings, treatment, and ultimately control of TB [49–51]. Thus, there is an urgent need to implement a TB awareness program which helps to enhance knowledge on TB and to reduce the social stigma and

**Table 3. Factors associated with knowledge of tuberculosis among people living with tuberculosis (PLTB).**

| Variables | Crude OR 95%(CI) | P-value | Adjusted OR (95% CI) | P-value |
|---|---|---|---|---|
| **Age in year** | | | | |
| 20–29 years | 1 | **< 0.001** | 1 | **< 0.001** |
| 30–39 years | 7.08 (3.32,15.09) | **< 0.001** | 6.49 (2.32,18.12) | **< 0.001** |
| 40–49 years | 6.87 (3.44,13.73) | **< 0.001** | 6.76 (2.85,16.05) | **< 0.001** |
| 50–60 years | 6.61 (2.89,15.13) | **< 0.001** | 13.96 (4.79,40.68) | **< 0.001** |
| **Gender** | | | | |
| Male | 1 | 0.022 | 1 | 0.736 |
| Female | 0.56 (0.34,0.92) | 0.023 | 0.89 (0.47,1.71) | 0.735 |
| **Marital** | | | | |
| Unmarried | 1 | 0.183 | 1 | 0.068 |
| Married | 1.49 (0.84,2.63) | 0.175 | 1.94 (0.96,3.9) | 0.065 |
| **Family** | | | | |
| Joint | 1 | **< 0.001** | 1 | **< 0.001** |
| Nuclear | 3.55 (1.94,6.51) | **< 0.001** | 3.92 (1.79,8.62) | **< 0.001** |
| **Ethnicity** | | | | |
| Janajati | 1 | 0.156 | 1 | 0.1 |
| Brahmin | 1.43 (0.87,2.36) | 0.156 | 1.69 (0.9,3.17) | 0.101 |
| **Education** | | | | |
| Illiterate | 1 | 0.892 | 1 | 0.431 |
| 1–7 class | 0.89 (0.53,1.49) | 0.653 | 1.71 (0.82,3.56) | 0.149 |
| 8–12 class | 0.79 (0.21,3.04) | 0.736 | 1.52 (0.27,8.48) | 0.635 |
| More than 12 class | 1.35 (0.37,4.86) | 0.647 | 2.25 (0.46,11.07) | 0.318 |
| **Occupation** | | | | |
| Labor | 1 | **< 0.001** | 1 | **< 0.001** |
| Agriculture | 1.24 (0.51,3.02) | 0.641 | 1.05 (0.39,2.83) | 0.927 |
| Private job | 5.4 (2.92,10.01) | **< 0.001** | 4.41 (2.11,9.21) | **< 0.001** |
| Government job | 4.04 (1.35,12.08) | **0.013** | 3.71 (1.04,13.15) | **0.043** |
| **Income** | | | | |
| Poor economic position | 1 | **0.003** | 1 | **_0.02_** |
| Middle economic position | 0.96 (0.57,1.62) | 0.865 | 1.45 (0.72,2.92) | 0.294 |
| High economic position | 3.91 (1.48,10.37) | **0.006** | 4.25 (1.41,12.87) | 0.01 |
| **Residence** | | | | |
| Urban | 1 | **0.002** | 1 | **_0.039_** |
| Rural | 0.46 (0.28,0.75) | **0.002** | 0.51 (0.27,0.97) 0.04 | **_0.04_** |

discrimination among all age groups of patients. The constitution of Nepal grants the right of every patient, and health counselling, and essential drugs, including anti-tuberculosis drugs, are also distributed free of cost through every public health facility [52, 53]. However, law, policy and health systems of Nepal, rural and even urban communities still face the issue of superstition which is causing an ongoing barrier to the acquisition of knowledge about TB [27].

Moreover, knowledge about TB was found to be significantly higher among patients from high economic positions and living in urban areas. Patients from urban areas were more likely to access health information and other general information, good communication, transportation and essential things. Subsequently, patients from high socioeconomic backgrounds had a significantly higher chance to satisfy their needs and demands more easily than patients poor socioeconomic positions. These findings indicate that patients from rural areas and poor

**Table 4. Factors associated with Utilization of DOTS among people living with tuberculosis (PLTB).**

| Variables | Crude OR 95%(CI) | P-value | Adjusted OR (95% CI) | P-value |
|---|---|---|---|---|
| **Age** | | | | |
| 20–29 years | 1 | < **0.001** | 1 | 0.179 |
| 30–39 years | 5.74 (2.79,11.84) | < **0.001** | 5.13 (2.03,12.96) | 0.179 |
| 40–49 years | 5.79 (2.96,11.32) | < **0.001** | 5.85 (2.64,12.96) | < **0.001** |
| 50–60 years | 5.82 (2.62,12.92) | < **0.001** | 10.84 (4.09,28.76) | < **0.001** |
| **Gender** | | | | |
| Male | 1 | 0.176 | 1 | 0.521 |
| Female | 0.72 (0.45,1.16) | 0.176 | 1.21 (0.67,2.18) | 0.522 |
| **Marital** | | | | |
| Unmarried | 1 | 0.163 | 1 | 0.08 |
| Married | 1.49 (0.86,2.57) | 0.156 | 1.78 (0.94,3.38) | 0.077 |
| **Family** | | | | |
| Joint | 1 | < **0.001** | 1 | **0.001** |
| Nuclear | 2.98 (1.73,5.15) | < **0.001** | 2.94 (1.49,5.82) | **0.002** |
| **Ethnicity** | | | | |
| Brahmin | 1 | 0.336 | 1 | 0.178 |
| Janajati | 1.26 (0.79,2.02) | 0.336 | 1.48 (0.84,2.6) | 0.179 |
| **Education** | | | | |
| Illiterate | 1 | 0.827 | 1 | 0.461 |
| 1–7 class | 0.95 (0.58,1.55) | 0.837 | 1.44 (0.76,2.73) | 0.266 |
| 8–12 class | 1.02 (0.26,3.8) | 0.321 | 1.84 (0.35,9.56) | 0.47 |
| More than 12 class | 1.7 (0.48,6.08) | 0.415 | 2.47 (0.54,11.28) | 0.243 |
| **Occupation** | | | | |
| Labor | 1 | < **0.001** | 1 | **0.017** |
| Agriculture | 1.27 (0.52,3.12) | 0.594 | 1.19 (0.45,3.15) | 0.729 |
| Private job | 3.08 (1.81,5.25) | < **0.001** | 2.39 (1.26,4.54) | **0.008** |
| Government job | 4.16 (1.39,12.44) | **0.011** | 3.66 (1.08,12.42) | **0.038** |
| **Income** | | | | |
| Poor economic position | 1 | **0.001** | 1 | ***0.006*** |
| Middle economic position | 0.84 (0.51,1.38) | *0.481* | 1.15 (0.61,2.15) | 0.666 |
| High economic position | 3.64 (1.48,8.97) | *0.005* | 4.2 (1.55,11.38) | 0.005 |
| **Residence** | | | | |
| Urban | 1 | **0.006** | 1 | ***0.05*** |
| Rural | 0.51 (0.32,0.83) | **0.006** | 0.57 (0.32,1) | 0.051 |

socioeconomic positions had no or inadequate access to basic fundamental requirements. The majority of poor patients faced the consequences of out-of-pocket payments. For instance, patients from poor socioeconomic backgrounds were unlikely to buy supplementary food, and other essential nutritious diets for combating the high doses anti-tuberculosis drugs [54]. Patients from poorer economic backgrounds were less likely to acquire adequate knowledge about TB [54]. Hence, these inequalities between poor and high income may cause a huge barrier to knowledge and also to diagnosis, control and elimination of TB. Similar findings were discussed in other literature, while persistent inequalities in socioeconomic positions have hampered the maintenance of patient care which lead to treatment failure [55–57].

On the other hand, 76.92% TB patients utilized the DOTS service from the public DOTS center while 23.07% of patients did not utilize the DOTS service. This higher percentage

(55.4%) who did not utilize the DOTS service was found to be in the 20 to 29 years age group. One possible reason behind this was poverty, along with the lack of access to roads and transportation, lack of knowledge on utilizing the DOTS, and discrimination (self and general public). Thus, to overcome these barriers to utilizing the DOTS service, the health authority and other stakeholders should pay more attention to alternative and decisive solutions, which can reduce the barrier and scale-up DOTS service utilization among the patients. Similar findings have also been mentioned in other literature, in particular lack of awareness on utilizing the DOTS on a daily basis, geographical condition, and social and economic position, which all play an important role in inhibiting the utilization of the DOTS service from the DOTS center [58–60].

Additionally, lack of competency of health workers, irregular opening hours of health facilities, and attitude of service providers are also causing barriers to utilization of the DOTS service from the public DOTS centers [27]. Thus, the government and other public private organizations should develop an appropriate strategy for introducing advanced medical equipment, providing in service training, developing updated guidelines, and focusing on early case findings. Other important issues are systematic screening of TB, strengthening the follow up mechanisms, proper monitoring and evaluation and supervision of TB patients, and organizing fixed opening hours of health care facilities. These strategies will help to control, prevent, diagnose, and treat TB in areas of Nepal [27, 61, 62].

The DOTS service was more likely to be utilized by those patients who had completed grade twelve, were working in private organizations, and/or who were living in urban areas. However, patients who were illiterate, living with a joint family and/or residing in rural areas were found to have a lower chance of utilizing the DOTS service from the public DOTS centers. These barriers were possibly due to the poor economic position, male dominated society, and lack of roads and transportation. To begin with, where people did not have qualifications, experience and skills in order to get a job in any organization, then they were less likely to earn good money. As a result, they were less able to afford basic requirements for their livelihoods. Further, low utilization of DOTS service was also found in patients who were living in a joint family. This is possibly due to the system of patriarchy of the country. In the majority of households in Nepal, every decision for utilizing health services and other essential issues is made by the head of the family which may hamper the utilization of the DOTS service. The findings of this study are consistent with previous literature that found that patients with no/low education, living in a joint family and residing in rural areas contribute to barriers to utilizing the DOTS service from the public DOTS center [27, 63, 64]. In summary, our study represents the important findings: (i) more than 20% of patients lacked adequate knowledge on TB; (ii) more than one quarter of TB patients did not utilize the DOTS service regularly; (iii) education, family type, income and residence were the main barriers for TB and utilization of the DOTS service. Hence, this finding indicates an urgent need for a TB program to enhance the knowledge and utilization of the DOTS service among TB patients.

Furthermore, the findings of this study had some limitations. Initially, we planned to carry out this study at Kathmandu Valley, however, due to the pandemic effect of COVID-19, the study is conducted in only one district (Lalitpur) of Kathmandu Valley. So, it might not be able to draw the best conclusion on knowledge of TB and utilization of DOTS among TB patients. Second, there might have a chance of recall bias and social desirability bias.

## Conclusion

This study investigates both the knowledge of TB and the utilization of the DOTS from the public DOTS centers in Lalitpur district, Nepal. It was found that knowledge of TB was found

to be significantly higher among the male patients and who were from 30–39 age groups. Similarly, income and mother's jobs were also found to increase knowledge on TB. Likewise, Utilization of the DOTS service from public DOTS centers was significantly higher among patients who were 50–60 years of age, living in a single family, have completed formal education and were living in rural areas. The findings of the study indicate that while knowledge of TB is satisfactory there is still a wide gap in the utilization of the DOTS service. Hence, appropriate programs should be implemented to fill the gap for utilization of the DOTS service from public DOTS centers.

## Supporting information

**S1 Data.**
(ZIP)

## Acknowledgments

The authors express gratitude to all respondents and enumerators. Despite the pandemic effect of COVID-19, they have provided immeasurable support for conducting this research. Additionally, the authors are thankful to the Institutional Review Committee of Novel College of Health Sciences, Sinamangal Kathmandu, Nepal.

## Author Contributions

**Conceptualization:** Nirmal Gautam.

**Data curation:** Nirmal Gautam, Rewati Raj Karki.

**Formal analysis:** Nirmal Gautam.

**Investigation:** Nirmal Gautam, Rewati Raj Karki.

**Methodology:** Nirmal Gautam.

**Resources:** Nirmal Gautam.

**Software:** Nirmal Gautam.

**Supervision:** Rasheda Khanam.

**Validation:** Rasheda Khanam.

**Visualization:** Nirmal Gautam.

**Writing – original draft:** Nirmal Gautam, Rasheda Khanam.

**Writing – review & editing:** Rasheda Khanam.

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
