## [Decision Letter · Decision Letter 0]

8 Nov 2020

PONE-D-20-31964

Knowledge on Tuberculosis and utilization of DOTS service by Tuberculosis patients in Lalitpur District, Nepal

PLOS ONE

Dear Dr. Gautam,

Thank you for submitting your manuscript to PLOS ONE. After careful consideration, we feel that it has merit but does not fully meet PLOS ONE’s publication criteria as it currently stands. Therefore, we invite you to submit a revised version of the manuscript that addresses the points raised during the review process.

We look forward to receiving your revised manuscript.

Kind regards,

Seyed Ehtesham Hasnain

Academic Editor

PLOS ONE

Journal Requirements:

2. Please include additional information regarding the survey or questionnaire used in the study and ensure that you have provided sufficient details that others could replicate the analyses. For instance, if you developed a questionnaire as part of this study and it is not under a copyright more restrictive than CC-BY, please include a copy, in both the original language and English, as Supporting Information, or include a citation if it has been published previously.

3. Please provide further details on sample size and power calculations.

4. In statistical methods, please clarify whether you corrected for multiple comparisons.

5. Please include your tables as part of your main manuscript and remove the individual files. Please note that supplementary tables (should remain/ be uploaded) as separate "supporting information" files

6.We note that you have indicated that data from this study are available upon request. PLOS only allows data to be available upon request if there are legal or ethical restrictions on sharing data publicly. For information on unacceptable data access restrictions, please see http://journals.plos.org/plosone/s/data-availability#loc-unacceptable-data-access-restrictions.

Additional Editor Comments (if provided):

Major Revision

Reviewers' comments:

Reviewer's Responses to Questions

**Comments to the Author**

1. Is the manuscript technically sound, and do the data support the conclusions?

Reviewer #1: No

Reviewer #2: Yes

2. Has the statistical analysis been performed appropriately and rigorously? 

Reviewer #1: N/A

Reviewer #2: Yes

3. Have the authors made all data underlying the findings in their manuscript fully available?

Reviewer #1: Yes

Reviewer #2: Yes

4. Is the manuscript presented in an intelligible fashion and written in standard English?

Reviewer #1: No

Reviewer #2: Yes

5. Review Comments to the Author

Reviewer #1: The content of the MS doesn't look sound enough to be a research article, it is a collection of some information and put it in one place systematically. Only with this content it does not match with the research article category of the PLOS ONE.

Reviewer #2: Comments:

Present study has been done by Gautam et al. and colleges were conducted very well on knowledge and understanding of TB disease and DOTS plus site among common population under NTP in Nepal. Following are some suggestions which can make the manuscript stronger and beneficial for the readers:

Abstract:

• Word limits excided as per Journal guidelines. Should be modified.

• .Write name of district (L. No. 26)

• L No. 32: 50-60 year age will be the higher age category. Author has advised to confirm the age category.

Introduction:

• Author has advised to shorten this section and put some point in to the discussion section.

Method:

• Author should specify the study duration (Date), instead of Mid- June 2020 to mid- August 2020 (L. No. 129-130)

• Ethical statement should write in separate sub-section.

Results:

• Author has advised to rewrite the results and specify the Table legends as according to the Journal guideline.

Discussion:

• An abbreviation of Tuberculosis (TB) and anti tuberculosis (anti-TB) were written anonymously by authors. Please correct.

• Authors should specify there limitation in present study.

• Conclusive statements should also modify to understand readers.

• Author has advised to use universal/ single system while writing the reference.

6. PLOS authors have the option to publish the peer review history of their article (what does this mean?). If published, this will include your full peer review and any attached files.

Reviewer #1: No

Reviewer #2: No

---

## [Author Response · Author response to Decision Letter 0]

26 Dec 2020

Editor: We have incorporated all of your suggestion. We are very thankful and would like to appreciated for your constructive suggestion

Reviewer 1: We have incorporated all of your suggestion. We are very thankful and would like to appreciated for your constructive suggestions. 

Reviewer 2: We have incorporated all of your suggestion. We are very thankful and would like to appreciated for your constructive suggestions.

---

## [Editor Report · Decision Letter 1]

6 Jan 2021

Knowledge on Tuberculosis and utilization of DOTS service by Tuberculosis patients in Lalitpur District, Nepal

PONE-D-20-31964R1

Dear Dr. Gautam,

We’re pleased to inform you that your manuscript has been judged scientifically suitable for publication and will be formally accepted for publication once it meets all outstanding technical requirements.

Kind regards,

Seyed Ehtesham Hasnain

Academic Editor

PLOS ONE

Additional Editor Comments (optional):

I have gone through the revised manuscript and also the Author response to the comments of the Reviewers. The authors have satisfactorily addressed the comments of both the reviewers and have revised the manuscript accordingly. Authors have updated the limitation of their study in revised manuscript . Authors have also made a few modifications to the conclusion of this study to make it further clearer. Introduction part has been shortened. All other explanations provided by the Authors to the comments of both the reviewers are quite satisfactory.

I recommend this manuscript for publication.

Decision : Accept
---

## [Editor Report · Acceptance letter]

14 Jan 2021

PONE-D-20-31964R1 

Knowledge on Tuberculosis and utilization of DOTS service by Tuberculosis patients in Lalitpur District, Nepal  

Dear Dr. Gautam:

I'm pleased to inform you that your manuscript has been deemed suitable for publication in PLOS ONE. Congratulations! Your manuscript is now with our production department. 

Kind regards, 

on behalf of

Prof Seyed Ehtesham Hasnain 

Academic Editor

PLOS ONE